# Equity of Health Services Utilisation and Expenditure among Urban and Rural Residents under Universal Health Coverage

**DOI:** 10.3390/ijerph18020593

**Published:** 2021-01-12

**Authors:** Jianqiang Xu, Juan Zheng, Lingzhong Xu, Hongtao Wu

**Affiliations:** 1School of Management, Xuzhou Medical University, Xuzhou 221004, China; 100002016020@xzhmu.edu.cn; 2School of Public Health, Shandong University, Jinan 250012, China; lzxu@sdu.edu.cn; 3School of Management, Tsinghua University, Beijing 100084, China; wuhongtao@tsinghua.edu.cn

**Keywords:** demand for health services, utilisation of health services, health expenditure, equity evaluation, urban-rural differences, universal health coverage

## Abstract

Worldwide countries are recognising the need for and significance of universal health coverage (UHC); however, health inequality continues to persist. This study evaluates the status and equity of residents’ demand for and utilisation of health services and expenditure by considering the three components of universal health coverage, urban-rural differences, and different income groups. Sample data from China’s Fifth Health Service Survey were analysed and the ‘five levels of income classification’ were used to classify people into income groups. This study used descriptive analysis and concentration index and concentration curve for equity evaluation. Statistically significant differences were found in the demand and utilisation of health services between urban and rural residents. Rural residents’ demand and utilisation of health services decreased with an increase in income and their health expenditure was higher than that of urban residents. Compared with middle- and high-income rural residents, middle- and lower-income rural residents faced higher hospitalisation expenses; and, compared with urban residents, equity in rural residents’ demand and utilisation of health services, and annual health and hospitalisation expenditures, were poorer. Thus, equity of health service utilisation and expenditure for urban and rural residents with different incomes remain problematic, requiring improved access and health policies.

## 1. Introduction

The World Health Organisation (WHO) regards quality, equity, and efficiency as the three major health system goals. Since 2015, an important objective of the global development agenda has been to achieve universal health coverage (UHC) [1]. UHC requires that everyone should have access to quality health services they require, without facing economic difficulties [2]. The connotations of this agenda are embodied in the following two aspects: equal access to national basic health services and economic risk protection against diseases [3]. Health financing aims to improve the health of the population, ensure the prevention of economic risk protection against diseases, and promote satisfaction of the health service needs of residents. The evaluation indicators of economic risk prevention in health financing include poverty caused by catastrophic health expenditure of households and personal cash expenditure. 

UHC and access for every citizen to equitable, accessible, and reliable health services and protection play an important and positive role in ensuring the political stability of a country, social security, and social equity. The concept of UHC has been around for a long time. In 1978, the Declaration of Almaty introduced ‘primary health care for all’, while, in 2005, the World Health Assembly proposed that everyone should have access to health services, without suffering financial hardships to pay for them [4,5]. The theme of the 2010 World Health Report was ‘Financing for Health Systems: The Road to Universal Coverage’. Subsequently, UHC was the focus topic of the World Health Report 2013 and World Health Day 2019 [6]. Today, promoting UHC has become a positive, politically-driven choice in most countries. By strengthening local investment in health and optimising health financing structures, middle-income countries, such as Brazil, Chile, China, and Mexico, have made significant progress in the past decade in terms of access to health services, reduction of out-of-pocket costs for health services, and efficiency and equity in the use of health resources [7].

While countries throughout the world are making progress, some 400 million people globally—one in 17—still lack access to basic health services. More people are forced to choose between health care and other daily expenses (e.g., food, clothing, or even a home). Therefore, the economic risk of diseases, wherein people must pay out-of-pocket to use health services, needs to be limited [8,9]. In 1980, a study group on health inequality in the United Kingdom published a report pointing out differences in morbidity, mortality, and use of health services among different segments of the population [10]. Studies have also shown that different income levels may lead to inequality in the economic disease burden [11]. Another study reported a bidirectional causal relationship between poverty and health, wherein poverty leads to poor health, which in turn deteriorates poverty levels [12].

As many countries have implemented the concept of UHC, its evaluation cannot be limited to the financing of health services [13]. It is also important to analyse the impact of funds raised on the use of health services by residents, especially at different economic levels, and to evaluate equity to systematically evaluate the impact of UHC on a country or region’s health system. The WHO stresses the importance of classifying measures according to socioeconomic and demographic ‘stratification’, in addition to measuring levels of coverage extended by basic health services and financial protection [14,15].

China’s health equity evaluation has attracted much attention from government departments and researchers due to its status as a developing country characterised with rapid social and economic growth and a large population. In August 2016, both the China Health and Wellness Conference and the outline of the ‘Healthy China 2030′ plan introduced the ‘strategic position of giving priority to people’s health and promoting the integration of health concept into the whole process of public policy formulation and implementation, through comprehensive, full-cycle health service delivery to achieve the goal of UHC’ [16,17]. Furthermore, at the 71st World Health Assembly in May 2018, China stated that UHC is a strategic priority for development [18]. 

Beginning in 2010, China has endeavoured to expand health coverage and provide its residents with fair access to primary health care with the intention of improving health equity [19]. To resolve the residents’ problem of ‘seeing a doctor is both difficult and expensive’ and ensure that everyone enjoys equal and accessible basic medical and health services, the Chinese government launched a new health care reform in 2009 [20]. In the subsequent 11 years (2009–2020), China’s health care reform has been implemented in four areas: basic medical service systems, public health service systems, basic medical security systems, and basic drug supply and security systems. The Chinese government has promoted the development and implementation of a universal medical insurance system, the reform of public hospitals, a hierarchical medical system, development of contracts with family doctors, and the basic drug system. This work covers the three dimensions of measurement of UHC realisation: population coverage, service coverage, and cost coverage [21]. Essentially, a ‘15-min medical treatment circle’ has been formed to reduce distance and time taken for residents to obtain health services [22]. To ensure that urban and rural residents enjoy equal access to basic medical care, China has established a medical security system covering 1.3 billion people, with a participation of over 95% [23]. The basic medical insurance system for urban workers has also been implemented, and the insurance premium is jointly paid by the unit and individual workers (accounting for 9% and 2% of the monthly salary, respectively). Meanwhile, for non-working urban and rural residents, the basic medical insurance for urban and rural residents has been made available. The basic medical insurance premium is jointly borne by individual contributions and financial subsidies. However, medical insurance is not compulsory for non-worker residents. The amount of individual contributions for preschool children and college students is appropriately reduced, and the individual payment of disabled people, orphans, widowed elderly, and other groups who have lost their ability to work is subsidised by the government [24]. China’s basic medical insurance ensures coverage for all residents by incorporating the fair practice of financing people with different ability to pay. The participation of all residents in the medical insurance system, expansion of the medical service items covered by insurance portfolios, and increase in the proportion of medical insurance compensation have all made health services more affordable. These developments are in line with the goals of UHC. 

Despite advances, China has experienced unique problems in the development of its health services. The urban-rural dual structure system is a significant contradiction that cannot be bypassed by China’s reform and development plan [25]. This system causes the allocation of resources, residents’ income, and consumption between urban and rural areas to be different and unequal. Some studies show that there are still considerable differences in health status, income, and health expenditure between urban and rural residents [26,27]. Although China has established a basic medical security system that covers its entire population, the traditional urban-rural dual structure makes it ‘fragmented’, leading to lack of efficiency and fairness. In addition, the basic medical insurance system suffers from considerable differences in coverage, financing level and mode, pooling level, and compensation policy, which further leads to a gap in financing, and medical service utilisation and benefit, unfavourable to achieving fairness. Thus, it is imperative to improve the equity of economic risk protection against diseases. The differences in residents’ ability to pay for health services caused by the differences between urban and rural structures and income is a problem requiring resolution as part of the process of achieving UHC. 

In the mid-1990s, Chinese scholars began to further explore the health status of urban and rural residents and the equity of health services. Several studies have shown that socioeconomic stratification of the population by income level, education level, and occupation is a primary factor behind discriminant utilisation of health services [28,29]. In China, multiple empirical studies have investigated the equity of health service utilisation among general residents, different insured groups, and different economically stratified groups [30,31]. However, in terms of research content, most scholars have focused on the method of measuring equity in health service utilisation [32,33], equity in health service utilisation, and its influencing factors [34,35,36]. These studies suggest a lack of debate and analysis of the equity of health service utilisation and health expenditure among urban and rural residents in the context of comprehensive health coverage in China. Moreover, in previous studies, the sample sizes considered were small, and limited empirical studies used large samples at the national or provincial level, thus making it difficult to comprehensively evaluate the fairness of health services.

The present study considered the three components of UHC (population, service, and cost coverage) along with health service needs, demand, and financing equity. Using data from the Fifth Health Service Survey of China (Shandong Province data), this study analysed the status of, and equity differences in, the demand for and utilisation of health services and health expenditure among urban and rural residents. Compared with previous studies, this research contributes to the existing literature in several ways. First, this study is based on China’s comprehensive health coverage and aims to explore urban and rural residents’ health service utilisation and health expenditure equity. Our findings can inform decisions on the formulation and optimisation of health policies related to the realisation of UHC. Moreover, this will improve accessibility to health services for all residents and the prevention of economic risks to help achieve UHC [37]. Second, this study used a large dataset, which not only ensures sample representativeness but also provides a reference for future research and enables comparison of the findings with those of the sixth National Health Service survey (data collection was completed in 2018, but the data have not yet been fully disclosed). Finally, the most important contribution of this study is the analysis of equity difference in health expenditures between urban and rural residents, a factor that very few studies have explored to date. Based on the vertical equity principle of ‘the poor pay less, and the rich pay more’ in health expenditure, this study used a novel approach, the findings of which may be considered as a reference for follow-up studies evaluating the equity of health expenditure. In these cases, equity is considered preferable if the concentration curve of residents’ health expenditure is below, and distant to, the 45° measurement line.

## 2. Materials and Methods

### 2.1. Study Sites and Participants

To ensure the samples’ representativeness, multi-stage, stratified cluster random sampling was utilised to identify survey sites and survey objects. In the first stage, counties/cities/districts were randomly selected in each of the 17 prefecture-level cities in Shandong Province. From second to fourth stages, this study randomly selected townships/streets (stage 2), villages/neighbourhood committees (stage 3), and households (stage 4). Finally, a total of 20 counties/cities/districts, 100 townships/streets, and 200 villages/neighbourhood committees were selected. The sampling was based on households whereby 60 to 65 households were randomly selected from each village/neighbourhood committee sample, based on the number of households in the location. Finally, a total of 12,006 valid households and 33,060 residents were investigated, including 6006 urban households (10,391 residents) and 6000 rural households (22,669 residents).

### 2.2. Data Collection

The Health Statistics Information Centre of the Health Department of Shandong Province organised and implemented the fifth National Health Service survey (part of Shandong Province), which involved a cross-sectional study. A face-to-face questionnaire interview was conducted with households and residents sampled using stratified, multi-stage cluster sampling from the entire province. The government compiled and adopted the ‘Household Health Questionnaire’ for this investigation. The questionnaire was answered by individuals who were most familiar with their household’s circumstances. The items in the questionnaire mainly concerned: (a) the general situation of the household (e.g., characteristics, income, expenditure, consumption expenditure, food consumption expenditure, and medical expenditure of the household); (b) personal information of household members (e.g., gender, age, educational level, health insurance, and marital status); (c) information on diseases, injuries, and utilisation of health services in the two weeks before the survey (e.g., whether they were sick, whether they were diagnosed with chronic diseases, and information on medical treatment undertaken and health expenditure incurred in the two weeks prior to the survey); (d) information on the utilisation of hospitalisation services in the year before the survey on household members (e.g., whether they were hospitalised and information on health expenditure incurred in the last year); (e) information on the health survey of children under five years of age; and (f) a health survey of women aged 15–64 years. After the on-site investigation, the data entry, cleaning, and analysis were jointly completed by researchers and masters or doctoral degree candidates at the School of Public Health of Shandong University. 

### 2.3. Indicator Calculation

The current situation and equity of residents’ demand for health services, utilisation of health services, and health expenditure incurred were evaluated from the perspectives of urban and rural areas and different income groups. Urban and rural categories were determined according to the nature of residents’ registered residences, which were divided into agricultural and non-agricultural groups. Different income groups were determined using the method of ‘five levels of income classification’ [38]. First, researchers investigated the extent of the urban households’ disposable income and rural households’ net income in the previous year. Then, households were divided into five levels from low to high household annual income. Residents’ demand for, and utilisation of, health services were measured by routine indexes including the prevalence rate of diseases in the two weeks prior to the survey, annual prevalence rate of chronic diseases, medical treatment rate in the two weeks prior to the survey, and annual hospitalisation rate. Medical and health expenditure was measured based on personal cash expenditure on outpatient services and hospitalisation. The health expenditure here refers to the expenses paid by residents after payment of medical insurance compensation (out-of-pocket expenses) (Table 1). 

### 2.4. Statistical Methods

The SPSS 22.0 statistical software package (IBM, Armonk, NY, USA) was used for variable transformation and statistical analyses. The counting data were expressed as relative numbers. An X2 test was used for comparison between the two groups, and the trend X2 test was used for comparison between multiple groups. The measurement data of non-normal distribution were expressed as the median (quartile range) and compared using non-parametric statistical tests. The comparison between the two groups was based on the medium Mann–Whitney U rank sum test, and the comparison between multiple groups was based on the Kruskal–Wallis H rank sum test. For evaluating the distribution and equity of residents’ prevalence of diseases in the two weeks prior to the survey, prevalence of chronic diseases, medical treatment rate in the two weeks prior to the survey, and annual hospitalisation rate in the last year for different income groups, the concentration curve and concentration index (CI) were used for demonstration and calculation [39]. The value of CI was determined as (−1, 1). The more the CI deviated from 0, the poorer the equity. If the CI was negative, the concentration curve was above the equity line, indicating that the occurrence of observation indicators was more concentrated in the low-income group. However, if CI was positive, the opposite was true [7]. When evaluating the distribution and equity of the annual medical and health, outpatient, and hospitalisation expenditures of households in different income groups, under the condition that the CI was (−1, 1), a positive concentration index indicates better equity. The concentration curves were all located below the 45° measurement line, indicating that the amount of medical and health expenditure was concentrated in the relatively high-income group; if the CI was negative, the opposite was true. This principle is based on the vertical equity principle of ‘low-income residents pay less, and high-income residents pay more’, in health economics (this is only for the reference of the vertical equity evaluation of health expenditure in this study). Differences were considered statistically significant at *p* < 0.05.

## 3. Results

### 3.1. Urban and Rural Residents’ Demand for and Utilisation of Health Services

The data of 12,006 households (6006 urban households and 6000 rural households) and 33,060 residents (10,391 urban residents and 22,669 rural residents) were obtained. These data show that there were statistically significant differences between urban and rural residents (*p* < 0.001) in the prevalence rate of diseases in the two weeks prior to the survey, prevalence rate of chronic diseases in the last year, and medical treatment rate in the two weeks prior to the survey. However, there was no statistically significant difference in the hospitalisation rate between urban and rural residents (*p* > 0.05) (Table 2).

The researchers compared the residents’ demand for and utilisation of health services in different income groups. They found that the prevalence rate of diseases in the two weeks before the survey, the prevalence rate of chronic diseases in the last year, the medical treatment rate in the two weeks before the survey, and the hospitalisation rate of rural residents in different income groups decreased with an increase in income; the difference was statistically significant (*p* < 0.001). There was no statistically significant difference in the prevalence rate of diseases in the two weeks before the survey, the prevalence of chronic diseases in the last year, the medical treatment rate in the two weeks before the survey, or the hospitalisation rate of urban residents in different income groups (*p* > 0.05) (Table 2).

### 3.2. Analysis of the Equity Difference of Demand for, and Utilisation of, Health Services between Urban and Rural Residents

The concentration curves of rural residents’ prevalence of diseases in the two weeks before the survey (CI = −0.1179) and prevalence of chronic diseases in the last year (CI = −0.1682) were above the equity line. This suggests that, for rural residents, these two indicators were concentrated in the low-income group. The concentration curves of prevalence of diseases in the two weeks prior to the survey (CI = 0.0129) and prevalence of chronic diseases in the past one year (CI = 0.0064) of urban residents were slightly below the equity line. This indicates that these two indicators for different income levels were slightly concentrated in the relatively high-income group. The absolute CI value of the prevalence of diseases in the two weeks before the survey and the prevalence of chronic diseases in the last one year among rural residents was greater than that of urban residents. This indicates that the equity of rural residents’ demand for health services was poorer than that of urban residents (Figure 1a,b).

The concentration curve of medical treatment undertaken in the two weeks prior to the survey for rural residents (CI = 0.0800) was below the equity line, while that of urban residents (CI = −0.0064) was above the equity line. This indicates that the equity of medical treatment undertaken in the two weeks prior to the survey for rural residents was poor and concentrated in the relatively high-income group. However, medical treatment secured in the two weeks before the survey for urban residents was relatively equal, with a slight tendency to distribution among the low-income group (Figure 1c). The concentration curves for the utilisation of hospitalisation services by rural residents (CI = −0.1124) and urban residents (CI = −0.0015) were above the equity line. This indicates that the utilisation of hospitalisation services by urban and rural residents was concentrated in the low-income group and that the equity of hospitalisation for rural residents was poorer than that for urban residents (Figure 1d).

### 3.3. Health Expenditure of Urban and Rural Residents

The annual household medical and health expenditure of rural residents was lower than that of urban residents. However, the expenditure for the two-weeks pre-survey outpatient services and annual hospitalisation costs of rural residents was higher than that of urban residents. The difference was statistically significant (*p* < 0.05). When comparing the health expenditure of residents in the different income groups, there was no statistically significant difference in the households’ annual medical and health expenditure and residents’ two-weeks pre-survey outpatient expenditure in different income groups in rural areas (*p* > 0.05). Compared with the residents in the middle- and high-income groups, residents in the middle- and low-income groups had a heavier burden of annual hospitalisation expenses; the difference was statistically significant (*p* < 0.05). There was no statistically significant difference in the two-weeks pre-survey outpatient expenditure of residents in different income groups in the urban areas (*p* > 0.05). Furthermore, the higher the household income, the higher was the households’ annual medical and health expenditure and residents’ annual hospitalisation expenditure (*p* < 0.05) (Table 3).

### 3.4. Analysis of the Equity Difference between Urban and Rural Health Expenditure

Based on the vertical equity principle of ‘the poor pay less, and the rich pay more’ in health expenditure, the description of this section is considered a reference for the equity evaluation of health expenditure. If the concentration curve of residents’ health expenditure is below and distant from the 45° measurement line, the equity is preferable. The study found that the concentration curve of rural households’ annual medical and health expenditure (CI = 0.0164) was only slightly below the 45° measurement line, almost intersecting with it, while the concentration curve of urban households’ annual medical and health expenditure (CI = 0.0921) was below the 45° measurement line. This shows that the equity of annual medical and health expenditure of urban households was preferable, while that of rural households was poorer (Figure 2a).

The results of the analysis of the equity difference between urban and rural residents’ two-weeks pre-survey outpatient health expenditure are as follows: the concentration curves of the outpatient health expenditure of rural (CI = −0.3933) and urban (CI = −0.0941) residents were above the 45° measurement line. This indicates that outpatient expenditure was concentrated in the relatively low-income group in urban and rural areas, with poor equity (Figure 2b).

The results of the analysis of the equity differences between urban and rural residents’ annual hospitalisation expenditure were as follows: the concentration curve of rural residents’ hospitalisation expenditure was distributed slightly above the 45° measurement line (CI = −0.0122). Moreover, hospitalisation expenditure was concentrated in the relatively low-income group, with poor equity. The concentration curve of hospitalisation expenditure of urban residents was below the equity line (CI = 0.0164), with good equity (Figure 2c).

## 4. Discussion

There are differences and inequity in health status, utilisation of health services, and health expenditures of urban and rural residents. These discrepancies may have been brought about by the urban-rural dual structure. The Chinese government is attempting to bridge these differences through the strategy of ‘integrated development of urban and rural areas’. This strategic goal requires improvement in the accuracy of policies. For this purpose, data from the National Health Service survey, conducted every five years, is collected and is representative since it is based on a large sample size. Thus, a reasonable analysis of these data is a good policy support tool.

This study explored the current situation and equity differences of health service needs, utilisation, and health expenditure of urban and rural residents from the perspective of different income groupings. The results are of great significance to attempt to achieve UHC in terms of coverage of health services, protection of economic risks of diseases, and improvement of accessibility of health services.

The results show that there were obvious differences in demand for and utilisation of health services between urban and rural residents (*p* < 0.001). Compared with urban residents, the prevalence rate of diseases in the two weeks before the survey and annual prevalence rate of chronic diseases in rural residents were lower, but the medical treatment rate in the two weeks before the survey was higher. There was no statistically significant difference in the annual hospitalisation rate between urban and rural residents (*p* > 0.05).

Why is the demand for health services low, but the utilisation of health services high, for rural residents? Some studies have argued that, since rural residents have lower health literacy, there are fewer residents undertaking self-diagnosis and treatment in rural areas compared to urban areas [40]. Some scholars also suggested that rural residents have better access to primary medical institutions in terms of time, distance, and cost due to improved access to medical services in these areas [41]. In recent years, the high utilisation rate of rural health services was the result of demand being met after the improvement of health service availability. However, studies have also suggested that basic medical services for rural residents are not yet being adequately provided compared to those for urban residents. The poorest quintile group was least likely to be provided with sufficient or advanced health care services compared with the wealthiest quintile group [42].

Rural residents’ health service needs and annual hospitalisation service utilisation were concentrated in low-income groups. The equity of these indicators for rural residents was poorer than that for urban residents. The demand for and utilisation of health services by different income groups in rural areas were statistically different (*p* < 0.001). The prevalence rate of diseases in the two weeks prior to the survey, the annual prevalence rate of chronic diseases, the medical treatment rate in the two weeks prior to the survey, and the annual hospitalisation rate of rural residents increased concomitant to a decrease in income. International evidence has corroborated these findings, showing that people with lower economic status are more likely to suffer from serious illness and may become further impoverished due to medical costs [43]. Another reason for this phenomenon is that diseases or hospitalisation not only limit the economic output of households by limiting an individual’s ability to work productively, but also account for higher healthcare expenditure [44,45]. This study did not find that the demand for and utilisation of health services of urban residents varied with their income (*p* > 0.05).

The results of this research showed that there were statistically significant differences in the annual medical and health expenditures, the two-weeks pre-survey outpatient expenditure, and the annual inpatient expenditure of urban and rural residents (*p* < 0.001). The annual household medical and health expenditure of rural residents was lower than that of urban residents. However, the expenditure for the two-weeks pre-survey outpatient services and annual hospitalisation costs of rural residents were higher than that of urban residents. At the time of the survey, the income of urban residents (28,264 yuan) was 2.75 times that of rural residents (10,260 yuan), but the expenditure on outpatient and hospitalisation was lower than that of rural residents, indicating that the burden of health expenditure on rural residents was relatively heavier. Considering the social and economic backgrounds linked to the higher income of urban residents and lower income of rural residents, the two-weeks pre-survey outpatient and annual hospitalisation expenditures in urban and rural areas probably led to this inequality. Thus, compared with urban residents, rural residents bear a heavier economic burden of disease.

Based on the vertical equity principle of ‘the poor pay less, and the rich pay more’ in health expenditure, our study found an inequality in the two-weeks pre-survey outpatient expenditure of urban residents, while the annual medical and health expenditure and hospitalisation expenditures were fairer. The equity of the health expenditure of rural residents was worse than that of urban residents. Furthermore, we compared the health expenditure of residents in different income groups, finding that the annual medical and health expenditure and the two-weeks pre-survey outpatient expenditure of low-income rural residents resulted in a higher economic burden of disease. The economic burden of annual medical and health expenditure and the two-weeks pre-survey outpatient expenditure of rural low-income residents was also relatively high. Moreover, the economic burden of annual hospitalisation expenditure of rural middle-income and low-income groups was relatively heavy. Some research has explained this phenomenon further: due to economic reasons, a larger proportion of low-income people in rural areas do not desire, or cannot, purchase basic medical insurance; therefore, these individuals need to pay higher out-of-pocket medical expenses when they seek treatment [32,46].

The core of UHC is to realise equal access to medical and health services, economic risk protection against disease, and enhanced availability and affordability of universal medical and health services. However, there is still a large gap in the allocation of health resources between urban and rural areas, and the financing criteria for low-income groups are not yet accurate. These factors lead to inequalities in access to and affordability of health services for urban and rural residents. Based on the objectives of UHC, this study puts forward policy suggestions to improve the accessibility of health services, prevent economic risks of diseases, and promote health equity.

First, this study considers improvement in the availability of basic medical and health services and narrowing of the differences in the need for, and utilisation of health services between urban and rural residents. For example, improvements in the universal accessibility of health services for residents through training and use of general practitioners in both rural and urban primary health care institutions are advisable. General practitioners can provide basic medical and public health services and play an important role in improving the population’s health literacy [47,48]. Second, the medical security system (insurance) and compensation policies should be refined to improve the universal affordability of medical and health services [49]. The existing basic social medical security system has benefited everyone, but there is no precise insurance level for low-income groups. Low-income individuals, especially rural residents, bear a heavy economic burden of disease in their use of health services. Therefore, the population’s receipt of medical insurance compensation needs to be subdivided. Moreover, the extent of health care benefits among different groups should be gradually balanced. This will effectively lead to ‘disease risk sharing’ by medical insurance and improve the affordability of medical care [50]. Finally, optimisation of health financing is an important component in implementing UHC. This would comprise tax revenue in the secondary distribution of financing and compensation among different income groups [51]. Health financing should design and implement standards for different income groups at the initial stage and reduce the level of financial burden for low-income groups as much as possible [52]. The proportion of government spending on health should be increased, while the proportion of personal spending on health should be reduced. This will ensure that low-income people benefit from the secondary distribution of health services in terms of supply and medical insurance reimbursement [53]. There is, of course, a very important but difficult task required in this regard: how to identify and classify low-income groups and those in urgent need of economic protection against disease risks [54]?

The findings of the current study should be considered in the context of several limitations. First, although the sample size of the data used in the study was large, the residents’ annual household income, family medical and health expenditure, two-weeks pre-survey medical treatment expenditure, annual hospitalisation expenditure, and other data were estimated based on resident reports. To an extent, these data reflect some objective facts, but residents may deliberately overestimate or underestimate their health expenditure due to potential perceived benefit factors. Second, the results need to be reviewed with caution, particularly for future replication. The sample population was from a relatively economically developed province in China. In addition, this study only describes and analyses the current status and equity of health service needs, utilisation, and health expenditure of urban and rural residents. However, it did not explore factors affecting these differences. Therefore, based on the analysis results of the equity differences, future studies should use centralised exponential decomposition or structural equation models to control confounding factors (e.g., age and disease), as well as conduct influence factor analysis and further in-depth research.

## 5. Conclusions

This study analyses the current status and equity differences of health service needs, utilisation, and health expenditure of urban and rural residents. The results showed that both the prevalence rate of diseases in the two weeks prior to the survey and annual prevalence rate of chronic diseases in rural residents were lower than those in urban residents. However, the medical treatment rate in the two weeks before the survey for rural residents was higher than that of urban residents. Rural residents’ prevalence of diseases in the two weeks before the survey, annual prevalence of chronic diseases, and annual hospitalisation service utilisation were all concentrated in the low-income group. The results of the centralised curve analysis showed that the equity of health service need and utilisation by rural residents was poorer than that of urban residents. The expenditure on two-weeks pre-survey outpatient services and annual hospitalisation for rural residents was higher than that for urban residents. The outpatient expenditure and hospitalisation expenditure were concentrated in relatively low-income groups in rural areas, with poor equity. The results of this study indicate that achieving equity in health service utilisation and expenditure over different income groups, particularly rural residents, is problematic and requires consideration. In the context of UHC, health is a fundamental human right, and equity is the core of the system. In rural areas, low-income groups should be the focus of implementation of UHC. In the future, efforts should be made to ensure accessibility of basic medical services, improve the utilisation of health services, and meet the needs of health services for low-income groups. To improve the equity of health services, government departments should implement effective measures to develop health financing policies which ensure coverage by medical insurance and the compensation of medical expenses. Policymakers should also promote the protection of economic risks of diseases for low-income groups. Thus, the goal of UHC can only be realised through collective efforts.

## Figures and Tables

**Figure 1 ijerph-18-00593-f001:**
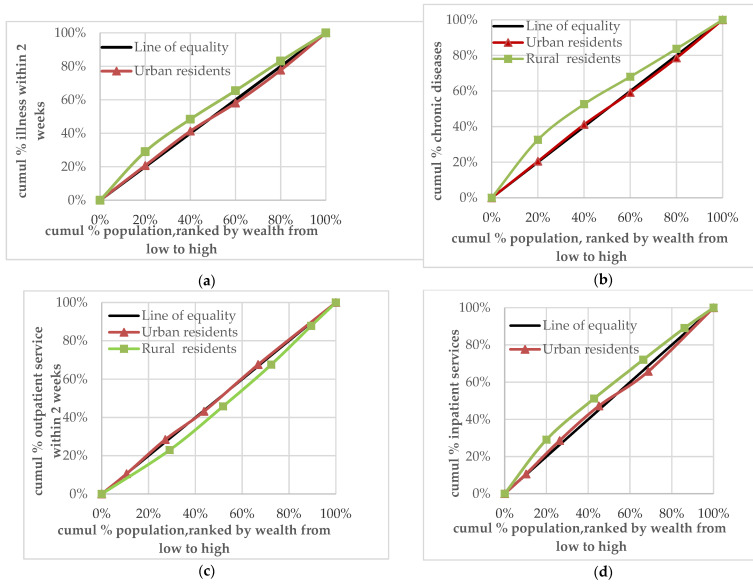
(**a**) Concentration curve of urban and rural residents’ two-week disease prevalence; (**b**) concentration curve of urban and rural residents’ chronic disease prevalence; (**c**) concentration curve of urban and rural residents’ outpatient service utilisation; (**d**) concentration curve of urban and rural residents’ inpatient service utilisation.

**Figure 2 ijerph-18-00593-f002:**
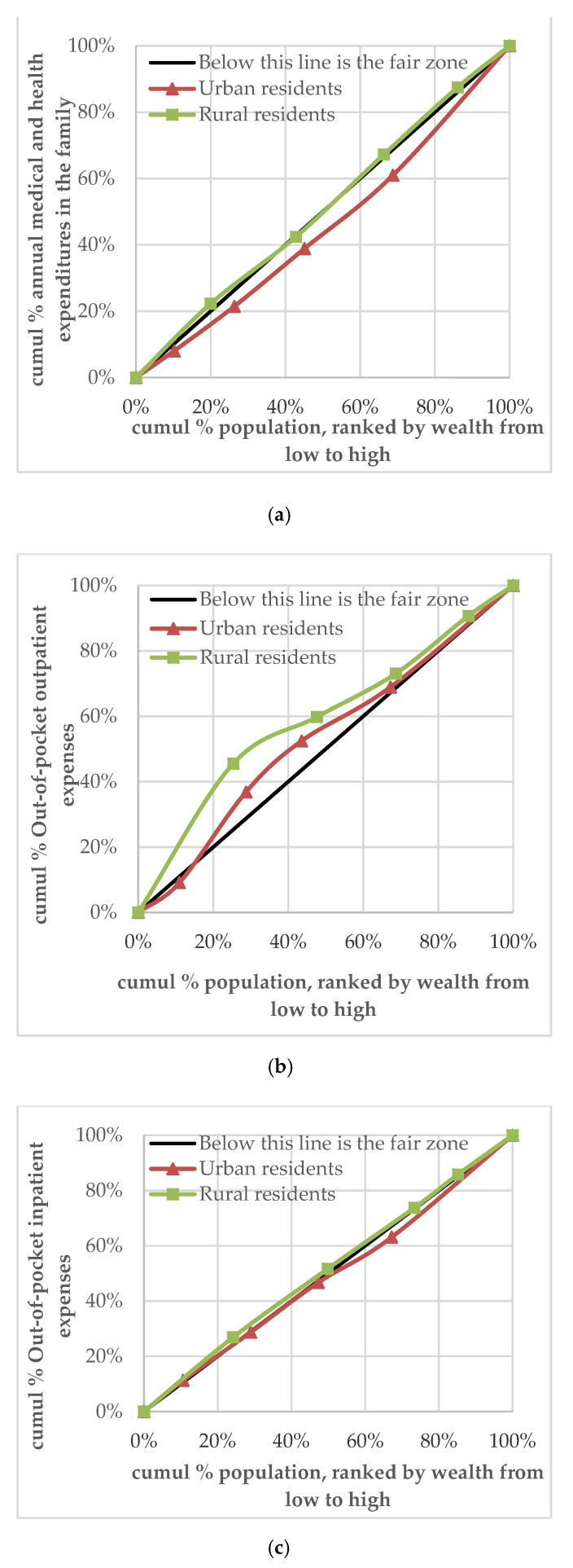
(**a**) Concentration curve of urban and rural households’ health expenditure; (**b**) concentration curve of urban and rural residents’ out-of-pocket payment for outpatient services; (**c**) concentration curve of urban and rural residents’ out-of-pocket payment for inpatient services.

**Table 1 ijerph-18-00593-t001:** Description of explanatory variables.

Variable	Survey Questions and Description
**Variable of basic characteristics of the population**	
Urban and rural categories	What is the nature of your registered residence? 1 = Agriculture; 2 = Non-agriculture
Income	How much was your household’s total income in the previous year? (Disposable income for urban households and net income for rural households)
Classification into 5 groups by income	Customised variables: Using the method of five levels of income classification, researchers divided urban and rural households into 5 levels according to their income. (Low-income group, lower middle-income group, middle-income group, upper middle-income group, and high-income group)
**Measurement variables of residents’ demand for and utilisation of health services**	
Getting sick in two weeks	Did you have any illness or injury in the two weeks before the investigation? 1 = Yes; 2 = No
Annual chronic disease	Have you been diagnosed with chronic diseases (hypertension, diabetes, cancer, and other chronic diseases) in the past year? 1 = Yes; 2 = No
Medical treatment in two weeks	Have you seen a doctor for a disease or injury in the last two weeks? 1 = Yes; 2 = No
Hospitalised	In the past 12 months, have you been diagnosed by a doctor as requiring hospitalisation but you have not been hospitalised? 1 = Yes; 2 = No
**Measurement variables of medical and health expenditure**	
Personal cash health expenditure on outpatient services	How much did you pay for your illness in the last two weeks? (Excluding reimbursement and expenses paid by personal medical accounts)
Personal cash health expenditure on hospitalisation	How much did you pay for your hospitalisation expenses in the last year? (Excluding reimbursement and expenses paid by personal medical accounts)

**Table 2 ijerph-18-00593-t002:** Comparisons of health service demand and utilisation among urban and rural residents with different income groups.

Urban or Rural/Income Group	No.	Getting Sick in Two Weeks (*n* %)	Annual Chronic Disease (*n* %)	Medical Treatment in Two Weeks (*n* %) ^a^	Hospitalised (*n* %)
**Urban and rural areas**					
Rural areas	22,669	5096 (22.48)	4972 (21.93)	1813 (8.00)	172 (7.60)
Urban areas	10,391	2656 (25.56)	2741 (26.38)	643 (6.20)	801 (7.70)
*χ2*		37.780	78.722	32.501	0.090
*p-*Value		0.000	0.000	0.000	0.773
**Rural households**					
Low-income group	4535	1478 (32.59)	1621 (35.74)	417 (9.20)	503 (11.10)
Lower middle-income group	5170	1162 (22.48)	1178 (22.79)	414 (8.00)	383 (7.40)
Middle-income group	5333	1048 (19.65)	940 (17.63)	395 (7.40)	357 (6.70)
Upper middle-income group	4477	870 (19.43)	739 (16.51)	367 (8.20)	295 (6.60)
High-income group	3154	537 (17.03)	494 (15.66)	221 (7.00)	189 (6.00)
*χ2* ^b^		281.059	548.474	8.728	72.491
*p-*Value		0.000	0.000	0.003	0.000
**Urban households**					
Low-income group	1064	282 (26.50)	288 (27.07)	68 (6.40)	85 (8.00)
Lower middle-income group	1669	439 (26.30)	454 (27.20)	115 (6.90)	144 (8.60)
Middle-income group	1953	439 (22.48)	476 (24.37)	95 (4.90)	148 (7.60)
Upper middle-income group	2459	614 (24.97)	632 (25.70)	157 (6.40)	150 (6.10)
High-income group	3246	882 (27.17)	891 (27.45)	209 (6.40)	273 (8.40)
*χ2* ^c^		0.927	0.164	0.019	0.153
*p-*Value		0.336	0.686	0.890	0.696

Note: **^a^** Medical treatment rate in the two weeks refers to the proportion of people who went to see a doctor for an illness within two weeks of the survey among the total number of people surveyed. This indicator reflects the utilisation of outpatient services by residents. **^b^*****^,^*****^c^** These indicate the Cochran-Arbitrage trend test.

**Table 3 ijerph-18-00593-t003:** Comparisons of health expenditure among rural and urban residents with different income groups (Median, M; Quartile Range, Q; Unit: Yuan).

Income Groups	Households’ Annual Medical and Health Expenditure		Outpatient Expenditure		Hospitalisation Expenditure	
	No.	M (Q)	No.	M (Q)	No.	M (Q)
**Urban and rural areas**						
Rural areas	6000	1000 (2000)	1126	50 (180)	1232	2500 (3789.25)
Urban areas	6006	1300 (2500)	1336	40 (137)	1296	2000 (4000)
*U*		118,197,131.5		673,924.5		709,269.5
*p-*Value		0.000		0.000		0.000
**Rural households**						
Low-income group	1200	1000 (2700)	219	60 (180)	282	2000 (3141)
Lower middle-income group	1200	1000 (1600)	230	60 (180)	252	2850 (3600)
Middle-income group	1200	1000 (1725)	220	50 (180)	236	2550 (3500)
Upper middle-income group	1200	1000 (2000)	255	45 (183)	218	2300 (1900)
High-income group	1200	1000 (2000)	202	45 (185)	244	2800 (4100)
*χ2*		4.151		5.036		14.411
*p-*Value		0.386		0.284		0.006
**Urban households**						
Low-income group	1201	1000 (2140)	270	35 (107)	305	1800 (4700)
Lower middle-income group	1201	1000 (2500)	298	33 (108)	275	1580 (3500)
Middle-income group	1201	1000 (2500)	269	50 (185)	272	2000 (4075)
Upper middle-income group	1201	1700 (3000)	269	40 (135)	225	2000 (3450)
High-income group	1202	2000 (4000)	230	50 (198)	219	2400 (4000)
*χ2*		172.689		2.778		16.72
*p-*Value		0.000		0.596		0.002

## Data Availability

The data used and/or analyzed during the current study are available from the corresponding author on request.

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
