# Peer review of "Equity of Health Services Utilisation and Expenditure among Urban and Rural Residents under Universal Health Coverage"

_ijerph, 2021, doi:10.3390/ijerph18020593_

Round 1

Reviewer 1 Report

This paper outlines a study aimed at understanding the demand and utilization of medical services among rural and urban residents. The data uses a sample of residents form the Shandong province, China, who were surveyed as part of China’s Fifth Health Service Survey. The study was particularly interested in the differences between rural and urban residents, and those from the lower compared to higher income groups.

This is an interesting study with large and representative sample. The study is promising. However, I have high difficulty following the paper and understanding the aims and the theoretical background of the study. These should be elucidated further in the manuscript.

Also, the manuscript would benefit from thorough proofreading for grammar and flow by a native English speaker. For example, the first sentence explaining the objective of the study can be broken up into a few sentences to make it easier for readers to follow. Also, the sentence “The statistical analysis methods include descriptive analysis, chi-square test” in line 19 should be written in past tense and would be better if it was rephased to say “The statistical methods employed in the study included a descriptive analysis and a chi-square test”. A proofreading by a native English speaker would greatly improve the readability of the paper, making it more digestible and accessible to a wider audience.

Abstract

The sentences in the abstract are generally lengthy and confusing. They seem to discuss many dimensions all at once. For example, in line 21, the authors bring up the dimension of “rural and urban areas”, and “low-income groups in urban areas” in the same sentence. The abstract will be more digestible if each sentence focuses on one key concept.

Some sentences are also grammatically inaccurate. For example, in line 26, “The health expenditure of the low-income groups in rural areas is not significantly different…” should be written in past tense (i.e. was not significantly different) as it refers specifically to the conclusion drawn from the study’s sample.

Lastly, there are some awkward phrases in the abstract, such as “has big demand” (line 29). The paper will benefit from proof reading by others to ensure that these awkward phrases are caught.

Introduction

There is also some awkward phrasing in the introduction as well. For example, line 55 “From the declaration of Almaty in 1978, “Primary health care for all”, to World Health…” is oddly phrased and I am unsure if authors mean to say that the concept of UHC has been around for many years. The following sentence (“from the theme of the 2010 World …. 2019: Universal health coverage” Line 57-60) is also oddly phrased and difficult to understand.

There are also some run-on sentences that would be more digestible if described in a few sentences. For example, the first sentence of the last paragraph (Line 120-216) span six lines. This makes it difficult for readers to follow.

Materials and methods

The materials and methods should have been written in past tense since the study has been completed. Currently, section 2.1 is written in present tense, while section 2.2 contains a mix of past and present tense, and section 2.4 is written in present tense.

Results

Similarly, the results section should be written in past tense. Currently, there is a mix of present tense (e.g. line 209, “show that there are”) and past tense (e.g. line 218 “the difference was statistically insignificant). This inconsistency is observed throughout the results section.

In line 231, the authors state to see statistical description for details, but also states that the data is omitted in the table. The authors can make this statement clearer as it is confusing as to where readers can find these details.

There is also some awkward phrasing in some of the sentences. For example, in line 248, the sentence “The higher the households’ income… was statistically significant” is oddly phrased, as the statistical significance is implied in the first half of the sentence.

Section 3.4 employs a few symbols (/) in Lines 256, 259 and 261 which are not very intuitive. The flow of this section would be better if these symbols are replaced with “or” or “and” instead.

The authors must revise the Figure 2 to improve its quality.

Discussion

There are numerous grammatical errors in the discussion. For example, in line 315, the word “on” should not be present in the phrase “in their study on measuring health equity”. There are several tenses errors as well. When referring to the current study’s sample data, the authors should remember to use past tense (e.g. Line 341 “Hospitalization services for rural residents are…” should be “Hospitalization services for rural residents were…” since the sentence refers to the sample data).

In lines 316-323, it is unclear if the authors are referring to the present study, Sun X et al.’s study or both. It would be good if this could be clearly stated.

There are also some generally unclear phrases. For example, in Line 258, the phrase “challenges in how to realize equity” is a bit odd. The flow might be better if authors wrote “challenges in realizing equity” instead. Another example is the phrase “all the people will not have differences” (Line 388).

On top of that, some sentences are poorly worded and difficult to comprehend. For example, in line 405, the sentence “The strategies to improve the protection on residents from disease economic burden: Health department can…” is unclear. It does not seem like that colon (:) is the right punctuation to use. The first half of the sentence also does not imply that the authors will be giving suggestions as to how to better protect residents from disease economic burdens. Hence, the sentence could use some restructuring.

Also, there are run-on sentences in the discussion. For example, the sentence “In particular… family doctors” in lines 369-374, is rather lengthy and would benefit from being split into multiple sentences instead.

Conclusion

The conclusion sounds prescriptive in nature. For example, Line 432, “the low-income group will be the focus of the work to achieve UHC in the future”. If the authors are intending to propose this as a suggestion, they should consider using the word “should” or “can” instead of “will”.

The phrase “From the perspectives of the urban and rural areas” is used quite often throughout the paper. It would be good if authors can find ways to rephrase this statement as it is rather repetitive. Sometimes, they are also used unnecessarily. For example, in Line 442 there is no need for the phrase in the sentence “From the perspectives of rural and urban areas… verifies that the urban-rural dual structure…” since it is stated again that the authors are referring to the urban-rural dual structure.

Author Response

Dear Professor,

    Please allow me to express my sincere thanks to you for your sincere and professional suggestions for revision, which is of great help to the improvement of our manuscript. These high-level amendments will greatly improve the quality of the manuscript. After carefully understanding your questions and suggestions, I tried my best to revise the manuscript. I hope the revised manuscript can get your approval and support. Of course, if the modification of the manuscript does not meet your requirements, please forgive and suggest further modification. Thank you very much.

The modifications made according to the suggestions are as follows:

1.Abstract

Point 1: The sentences in the abstract are generally lengthy and confusing; some sentences are also grammatically inaccurate; there are some awkward phrases in the abstract.

Response 1: The writer has changed the long sentences in the abstract to short ones. It also fixed incorrect grammar and awkward phrases. Basically, the abstract was rewritten based on the revision of the research results and conclusions.

2.Introduction

Point 2:  There is also some awkward phrasing in the introduction as well; there are also some run-on sentences that would be more digestible if described in a few sentences.

Response 2:  There are also awkward phrases and some run-on sentences in the introduction. Therefore, the author sought help from professionals (Editage) for English language editing. Many confusing sentences have been modified to make the manuscript easier for the reader to understand.

3.Materials and methods

Point 3:  The materials and methods should have been written in past tense.

Response 3:The tenses in the article have been corrected by professionals. In addition, two variables (reasons for not seeing a doctor; reasons for not being hospitalized) were deleted from Table 1 because the results of the study were modified.

4.Results

Point 4:  Similarly, the results section should be written in past tense; in line 231, the authors’ state to see statistical description for details, but also states that the data is omitted in the table. as it is confusing as to where readers can find these details; there is also some awkward phrasing in some of the sentences; Section 3.4 employs a few symbols (/) in Lines 256, 259 and 261 which are not very intuitive; the authors must revise the Figure 2 to improve its quality.

Response 4: Similarly, the tenses and some awkward phrasing have been corrected by professionals. The results of 3.2(reasons why urban and rural residents did not see a doctor or be hospitalized in the two weeks) were deleted. The content of this part had little relevance with the main research content of this study. This study mainly analyzed the current situation and equity of urban and rural residents' health service needs, utilization and health expenditure. So the author deleted this part after much consideration. In section 3.4,the author deleted the symbol(/),and replaced it with “and”. Also, Figures 1 and 2 have been modified to make them more intuitive and clear. In addition, the results of this study were further improved. The order of the four parts of the results has been tweaked to suit the needs of the study.

5.Discussion

Point 5: There are numerous grammatical errors in the discussion; there are several tenses errors as well; there are also some generally unclear phrases; on top of that, some sentences are poorly worded and difficult to comprehend; also, there are run-on sentences in the discussion.

Response 5: The discussion section was rewritten based on the results of this study. The discussion section in the first draft was too sketchy and not closely related to the research results. And there were numerous grammatical errors, several tenses errors, unclear phrases and run-on sentences, which left the reader confused and unable to understand the research. So the author spent a lot of time revising the discussion section. The revised discussion was basically a detailed elaboration of each part of the research results. In addition, the grammatical errors, tenses, unclear phrases and run-on sentences have been corrected by professionals.

6.Conclusion

Point 6:  The conclusion sounds prescriptive in nature; the phrase “From the perspectives of the urban and rural areas” is used quite often throughout the paper.

Response 6: The conclusion of this study was also rewritten and it is a summary based on the revised research results and discussions. The prescriptive conclusion has been modified. And the phrase “From the perspectives of the urban and rural areas” has also been modified.

Reviewer 2 Report

Dear authors,

Thank you very much for sending your article to the journal. I think your article is of relevant interest, in the attached pdf you will find some indications.

Regards.

Author Response

Dear Professor,

    Please allow me to express my sincere thanks to you for your sincere and professional suggestions for revision, which is of great help to the improvement of our manuscript. These high-level amendments will greatly improve the quality of the manuscript. After carefully understanding your questions and suggestions, I tried my best to revise the manuscript. I hope the revised manuscript can get your approval and support. Of course, if the modification of the manuscript does not meet your requirements, please forgive and suggest further modification. Thank you very much.

The modifications made according to the suggestions are as follows:

Point 1:  The abstract should be more concrete.

Response 1: The abstract was rewritten based on the revision of the research results and conclusions. And it was more concrete than previous manuscript.

Point2:  The percentage and "n" symbols should appear in the table.

Response 2: According to the modification suggestion, the percentage and "n" symbols had been put into the table 2.

Point3:  The legend of the tables should collect more information, as the tables would look more like them.

Response 3: The legend of the tables had also been modified according to the modification suggestions. We explained an important indicator in the table. The indicator reflects the utilisation of outpatient services by residents.

Reviewer 3 Report

Thank you for the opportunity to review this study.

In my view, the authors are making too many vague arguments that the findings cannot suggest.

For example, "The health expenditure of the low-income groups in rural areas is not significantly different from that of other income groups in rural
areas, which is unequal." 

-> The authors are making too much argument with just simple descriptive analysis.

Many studies of healthcare expenditure research have great limitations, because there are few data in which healthcare costs were objectively measured. In addition, health insurance coverage will also be related to medical expenses, and in rural cities with a relatively large elderly population, medical expenses may be higher.

Also, were unmet needs care and catastrophic health expenditure considered in the study?

Without sophisticated control and analysis of too many factors of consideration, the authors are making ambiguous arguments.

Author Response

Dear Professor,

    Please allow me to express my sincere thanks to you for your sincere and professional suggestions for revision, which is of great help to the improvement of our manuscript. These high-level amendments will greatly improve the quality of the manuscript. After carefully understanding your questions and suggestions, I tried my best to revise the manuscript. I hope the revised manuscript can get your approval and support. Of course, if the modification of the manuscript does not meet your requirements, please forgive and suggest further modification. Thank you very much.

The modifications made according to the suggestions are as follows:

Point 1: The authors are making too much argument with just simple descriptive analysis.

Response 1: The question raised is very meaningful. It helped me think deeply about the whole study. The results of this study mainly discussed and analyzed the current situation and equity differences of health service needs, utilization and health expenditure of urban and rural residents and different groups. Therefore, the discussion and conclusion of this study should also focus on the research results. However, many of the discussions and conclusions in the previous manuscript deviated from the results of this study. As a result, our authors were making too much argument with just simple descriptive analysis. After recognizing this problem, the author made great efforts to rewrite the discussion and conclusion based on the results. Therefore, the rewritten discussion and conclusion was basically a detailed elaboration of each part of the research results. We hope that the modifications will make the study more readable. Thank you again for your valuable advice. Without your guidance, the quality of this study would not have been improved so much.

Point 2: Were unmet needs care and catastrophic health expenditure considered in the study?

Response 2: Dear Professor, I would like to thank you again for your important question on health economics research. Our survey includes information on the reasons why residents do not seek medical treatment or stay in hospitals, as well as household annual income and expenditure data, which support future measurements of catastrophic household health expenditures. Due to the limitation of research design and manuscript length, this study only analyzes the current situation and equity of residents' health service needs, utilization and health expenditure. In this study, residents' health expenditure includes two weeks' medical treatment expenditure, annual hospitalization expenditure, and total annual medical and health expenditures in the family. The study did not take into account unmet care needs and catastrophic health expenditures. Catastrophic health expenditures are the focus of further research. Further research will complete the measurement and analysis of the incidence, intensity, relative disparities, and equity of catastrophic health expenditures in the surveyed households.

Point 3: Without sophisticated control and analysis of too many factors of consideration, the authors are making ambiguous arguments.

Response 3: The question raised is also very meaningful. It helps me to think about what further research should focus on in the future. This study simply described the current situation and equity differences of health service needs, utilization and health expenditure of urban and rural residents and different groups. The description of equity only analyzed the differences of the residents' health service needs, utilization and health expenditure under the economic level. But it didn't analyze what were the factors that affect fairness. If we want to further analyze the factors that affect fairness, we must consider the influence of confounding factors, and we can use the method of the decomposition of the concentration index to study the contribution degree of different factors (such as age, insurance, etc.) to fairness. This will be the focus of future research. However, this study was only a description of the current situation and the differences in fairness. Therefore, in the limitation part of the discussion, the author had mentioned the limitations of this study and the prospect of future research.

Round 2

Reviewer 1 Report

The authors have sufficiently addressed my comments. There is significant improvement in the current manuscript. But I have a minor comment. I hope that the authors can improve their conclusion further. It can be revised to be more objective. Also, the writing is still awkward and has room for improvement. The research gap and the current contribution can be elaborated further.

Author Response

Dear Professor,

    Thank you very much for your valuable suggestions again, which will further improve our manuscript and make it more readable. After carefully understanding your questions and suggestions, I tried my best to revise the manuscript. I hope the revised manuscript can get your approval and support. Thank you very much.

The modifications made according to the suggestions are as follows:

Point 1: The conclusion can be improved further. It can be revised to be more objective.

Response 1: According to the modification suggestions, we have made appropriate modifications to the research conclusions. Most of the conclusions are based on the research results, which are concise and objective to the main research results. At the end of the conclusion, we give a brief countermeasure suggestion based on the problems found in the research. Further emphasis was given to addressing inequities in the use of health services and health expenditure for urban and rural residents in the context of universal health coverage. See lines 455 through 468 for the changes.

Point 2: The writing is still awkward and has room for improvement.

Response 2:  Because our English level is very limited, there will inevitably be some awkward sentences. In order to solve this problem, we again turned to professional language polishing agencies for help. Editage (www.editage.cn) is a professional language polishing agency. We hope that after further language polishing, this manuscript will be appreciated for your support and approval.

Point 3: The research gap and the current contribution can be elaborated further.

Response 3: The previous manuscript was not perfect in terms of research gap and research contribution. Therefore, we are very grateful for your valuable suggestion, which can help our manuscript to get a greater improvement. Modifications to this section are as follows: First of all, through consulting more literature and reviewing, we found that the current research mainly focused on the measurement method of equity in health service utilization, equity in health service utilization and its influencing factors, and so on. There is still a lack of discussion and analysis on the equity of health service utilisation and health expenditure of urban and rural residents in the context of comprehensive health coverage in China. Moreover, in the existing studies, the selected range of research objects is relatively narrow, and there are few large sample empirical studies at the national or provincial level. Therefore, according to the current research status, the research gap and contribution of this research mainly include the following three aspects: First of all, this study considers the three components of UHC (population, service, and cost coverage) to explore the urban and rural residents’ health service utilisation and health expenditure equity. Second, the data in this study are from large samples with good representativeness. Thirdly, few previous studies have analysed the equity of health spending. Based on the vertical equity principle of ‘the poor pay less, and the rich pay more’ in health expenditure, this study is considered as a reference for the equity evaluation of health expenditure, which can provide reference for the follow-up research. See lines 111 through 143 for the changes.